# GUIDED: Granular Understanding via Identification, Detection, and Discrimination for Fine-Grained Open-Vocabulary Object Detection

**Jiaming Li**[1,2*†]  **Zhijia Liang**[1*]   **Weikai Chen**[‡]  **Lin Ma**[2]   **Guanbin Li** [1,3,4§]

[1]School of Computer Science and Engineering, Sun Yat-sen University, Guangzhou, China
[2]Meituan, Shenzhen, China
[3]GuangDong Province Key Laboratory of Information Security Technology, China
[4]Research Institute, Sun Yat-sen University, Shenzhen, China

## Abstract

Fine-grained open-vocabulary object detection (FG-OVD) aims to detect novel object categories described by attribute-rich texts. While existing open-vocabulary detectors show promise at the base-category level, they underperform in fine-grained settings due to the semantic entanglement of subjects and attributes in pretrained vision-language model (VLM) embeddings – leading to over-representation of attributes, mislocalization, and semantic drift in embedding space. We propose GUIDED, a decomposition framework specifically designed to address the semantic entanglement between subjects and attributes in fine-grained prompts. By separating object localization and fine-grained recognition into distinct pathways, GUIDED aligns each subtask with the module best suited for its respective roles. Specifically, given a fine-grained class name, we first use a language model to extract a coarse-grained subject and its descriptive attributes. Then the detector is guided solely by the subject embedding, ensuring stable localization unaffected by irrelevant or overrepresented attributes. To selectively retain helpful attributes, we introduce an attribute embedding fusion module that incorporates attribute information into detection queries in an attention-based manner. This mitigates over-representation while preserving discriminative power. Finally, a region-level attribute discrimination module compares each detected region against full fine-grained class names using a refined vision-language model with a projection head for improved alignment. Extensive experiments on FG-OVD and 3F-OVD benchmarks show that GUIDED achieves new state-of-the-art results, demonstrating the benefits of disentangled modeling and modular optimization. Our code will be released in `https://github.com/lijm48/GUIDED`.

## 1   Introduction

Open-vocabulary object detection (OVD) offers greater flexibility than traditional closed-set detection by allowing models to recognize arbitrary categories specified by text prompts. This paradigm significantly improves scalability in real-world environments where new categories frequently emerge and manual annotation is costly. However, most existing OVD methods focus on coarse-grained

---

[*]Equally-contributed authors.

[†]Work done during an internship at Meituan.

[‡]This paper solely reflects the author's personal research and is not associated with the author's affiliated institution.

[§]Corresponding author.

39th Conference on Neural Information Processing Systems (NeurIPS 2025).

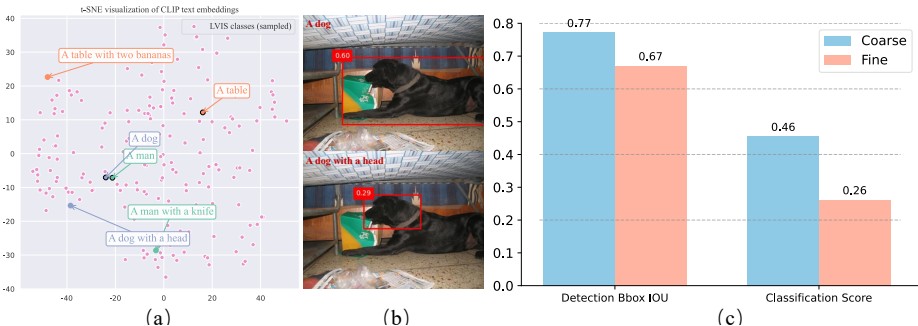

Figure 1: (a) The t-SNE visualization of CLIP text embeddings on LVIS classes and the fine-grained classes. The figure shows that some CLIP embeddings of fine-grained variants are positioned far apart. (b) The visualization of predictions of an OV detector with different class prompts(A dog vs A dog with a head). The detector focuses on the head instead of the dog, leading to incorrect localization. (c) The mean classification scores and the mean IoU of the prediction box with the ground truth box of OWL-ViT[24] under coarse-grained class queries and fine-grained class queries. The detector shows better performance on both classification and localization with coarse-grained queries than with fine-grained queries.

concepts (e.g. "dog", "cat") and fall short when dealing with more specific descriptions. Fine-grained open-vocabulary detection (FG-OVD) addresses this limitation by enabling recognition of novel categories with detailed attributes (e.g. "a small brown dog"). This fine-grained capability is vital for applications requiring precise granular understanding, such as product retrieval, visual search, and autonomous systems. Nonetheless, the increased semantic complexity in FG-OVD introduces new challenges in aligning textual attributes with visual regions, making it a critical yet under-explored problem in the open-vocabulary setting.

Existing FG-OVD methods typically rely on pretrained vision-language models (VLMs), such as CLIP, by directly encoding the full fine-grained class name into a single text embedding. However, this paradigm introduces two fundamental issues. First, due to the contrastive learning objective of VLMs, all tokens are treated equally, which leads to **semantic entanglement** between subjects and attributes. This often causes *attribute over-representation*, where descriptive modifiers dominate the embedding and suppress the core semantics of the object category. As shown in Figure 1(b), the query "a dog with a head" leads the model to focus only on the head, yielding mislocalized predictions, while "a dog" correctly grounds the entire object. Second, this entanglement also results in **semantic drift** in the embedding space. As illustrated in Figure 1(a), fine-grained variants like "a dog" and "a dog with a head" are positioned far apart in CLIP's latent space, despite their visually overlapping concepts. This mismatch makes classification unreliable for fine-grained open-set detection.

These issues stem from a common root: the use of **a single text embedding** to simultaneously **serve two objectives – object localization and attribute recognition**. Such coupling introduces semantic ambiguity, impeding the model's ability to specialize in either task. To address this, we propose GUIDED, a decomposition framework that disentangles FG-OVD into **coarse-grained object detection** and **fine-grained attribute discrimination**, allowing each to be handled by the model best suited for the subtask. This strategy is motivated by empirical observations shown in Figure 1(c): detectors achieve higher classification scores and more accurate localization when queried with coarse-grained categories (e.g., "dog") than with fine-grained descriptions (e.g., "a black fluffy dog"). This indicates that object detectors are better suited for base-level semantics, while fine-grained attribute recognition, which often requires subtle and localized reasoning, is more effectively handled by pretrained vision-language models.

To instantiate the design, GUIDED adopts a three-stage pipeline that explicitly separates subject identification, object detection, and attribute discrimination. Given a fine-grained class prompt, a large language model is first employed to extract the coarse-grained subject and its associated attributes, which are then encoded separately using a vision-language model. The subject embedding guides a coarse-grained object detector to localize candidate regions. To retain relevant attribute cues while avoiding over-representation, an attribute embedding fusion module selectively integrates

attribute embeddings into the detector queries via attention. In the final stage, fine-grained attribute discrimination is performed on the detected regions using region-text similarity. A lightweight projection head is applied to refine the text embeddings before comparison, enhancing the alignment between visual regions and fine-grained semantics. The final prediction score is computed by fusing the detector's coarse confidence with the attribute similarity score, yielding more precise and interpretable fine-grained predictions.

Extensive experiments on FG-OVD benchmarks validate the effectiveness of our proposed GUIDED framework, which outperforms existing state-of-the-art methods by a margin of $19.8\%$. Our main contributions are summarized as follows:

- We propose GUIDED, a novel decomposition framework that decouples FG-OVD into coarse-grained object detection and fine-grained attribute discrimination, aligning each subtask with the strengths of detection transformers and pretrained vision-language models.

- We design an attribute embedding fusion module that selectively integrates fine-grained attribute cues into detection queries, enhancing representation without overwhelming coarse category semantics.

- We introduce a projection-based attribute discrimination mechanism that refines text embeddings and computes region-text similarity for accurate fine-grained classification over detected objects.

- We establish new state-of-the-art results on FG-OVD benchmarks, demonstrating the effectiveness of task decomposition and modular optimization.

## 2 Related Work

**Open vocabulary object detection** Open-vocabulary object detection (OVD) has emerged as a salient research direction, propelled by advancements in vision-language models[25, 18, 13] and large-scale pretraining techniques. Notably, recent attempts [34, 24, 4, 35, 40, 15, 38, 37, 31] have primarily focused on adapting the VLMs to the detection task by fine-tuning. Another line of work [14, 32, 16, 39, 6, 12, 30] explores knowledge distillation to bridge the modality gap between detection and language understanding. ViLD [8] introduces a vision-language distillation framework that aligns region-level features with the image encoder from CLIP[25], thus enhancing cross-modal retrieval and detection accuracy. The recent works [33, 27, 36] integrate detection transformers in OVD to achieve further advanced capabilities. Grounding DINO[20] integrates the DINO with language models, enabling zero-shot object detection through text prompts by aligning visual regions with semantic embeddings. LAMI-DETR [5] introduces the language model instructions to generate the relationships between visual concepts for detection transformers. Despite these advances, existing methods exhibit limited performance on the detection of fine-grained classes(FG-OVD) with specific attributes due to the lack of fine-grained text-region annotations. Our approach addresses this limitation through decomposed FG-OVD into coarse-grained object detection with transformer detectors and fine-grained attribute identification with VLMs to take the inherent advantage of each model.

**Fine-grained Open Vocabulary Object Detection** The concept of fine-grained open-vocabulary detection (FG-OVD) [2] extends conventional OVD by introducing attribute-conditioned class definitions (e.g., color, material, shape) that require detectors to recognize novel classes like "dark brown wooden lamp" versus "gray metal lamp". Current approaches [28] predominantly address this challenge through text embedding refinement. SHiNe[19] proposes to update the classifiers in OVD by the hierarchy-aware sentences. However, it fails to capture the attribute information in its embedding construction process. HA-FGOVD[23] proposes a universal approach to generate attribute-highlighted text embeddings by masking the attention map of VLMs to obtain the attribute-specific features, while Bianchi et al.[1] propose to fine-tune an additional linear projection layer to enhance the fine-grained capability of CLIP text embeddings. However, these methods suffer from the attribute over-representation and semantic entanglement. Besides, the performance improvement led by embedding augmentation is limited by the detector's fine-grained capability. To address this, GUIDED addresses these limitations by decoupling attribute identification from detection pipelines, which integrates the strong discrimination capability of VLMs.

.

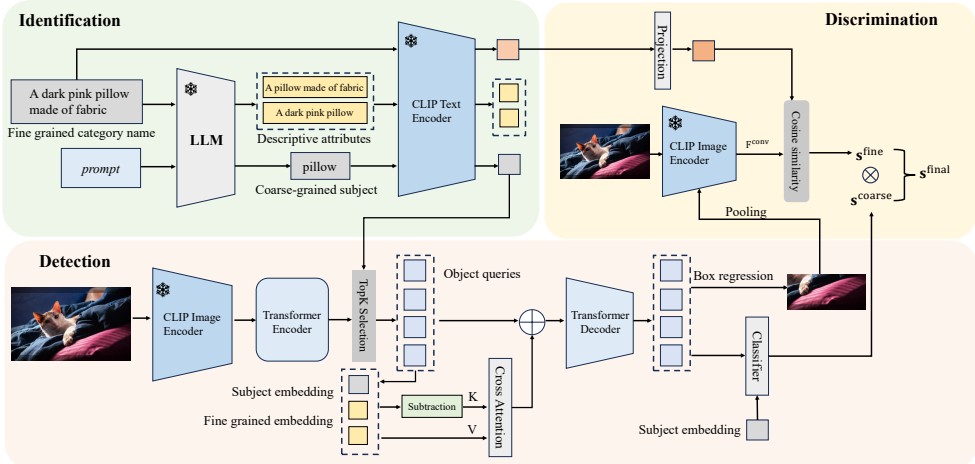

Figure 2: An overview of the proposed GUIDED framework. GUIDED adopts a three-stage pipeline, which consists of subject identification, coarse-grained object detection, and fine-grained attribute discrimination. In subject identification, GUIDED employs an LLM to extract the coarse-grained subject and its attribute embeddings. For detection, coarse-grained subject embeddings are adopted as queries to localize the candidate regions with coarse confidence. An attribute embedding fusion module selectively integrates attribute embeddings into queries. In the discrimination stage, GUIDED estimates the fine-grained score for each detected region with the full fine-grained class names using a refined CLIP with a projection head. The final score is obtained from the weighted multiplication of the detector's coarse confidence and the attribute similarity scores.

## 3 Approach

In this paper, we propose GUIDED, a framework specially designed for FG-OVD that aims to detect objects from both base and novel fine-grained categories with detailed attributes. Our proposed GUIDED framework disentangles the FG-OVD into different tasks, including subject identification, coarse-grained object detection, and fine-grained attribute discrimination. During the subject identification process, GUIDED introduces a large language model to identify the coarse-grained subject and fine-grained descriptions to generate the corresponding embeddings. In the coarse-grained object detection stage, a detection transformer is employed to localize objects based on subject embeddings while dynamically fusing attribute-specific semantics through an attribute embedding fusion module. Subsequently, fine-grained attribute discrimination adopts the VLMs to estimate the fine-grained scores on the detected proposal with the fine-grained text of each class. The overview of the GUIDED framework is shown in Figure 2.

### 3.1 Subject Identification

The proposed GUIDED framework addresses FG-OVD through a hierarchical decomposition strategy that systematically separates coarse-grained object detection and fine-grained attribute discrimination. This approach begins with semantic parsing of fine-grained class names using a frozen large language model. Given a fine-grained class name, we first identify its subject as a coarse-grained class and the associated attributes by prompting the existing large language models(e.g. GPT4-o [9]). These prompts are shown in Figure 3 (a). For each class, the identified subjects and associated attributes are fed into the frozen CLIP text encoder to obtain the coarse-grained text embeddings $\{\mathbf{t}_i\}_{i=1}^{n}$ and the attribute embeddings $\{\{\mathbf{t}_i^j\}_{j=1}^{n_j}\}_{i=1}^{n}$. Here $n$ is the number of classes and $n_j$ is the number of attribute embeddings for the $j$-th class. Note that the coarse-grained Subject Identification is done before the training or inference process.

The coarse-grained subject identification process can also be applied to identify the super class of a subclass, such as identifying the dog from the Siberian Husky. In this case, the associated attributes can be extended from the descriptions of the complex classes by LLMs.

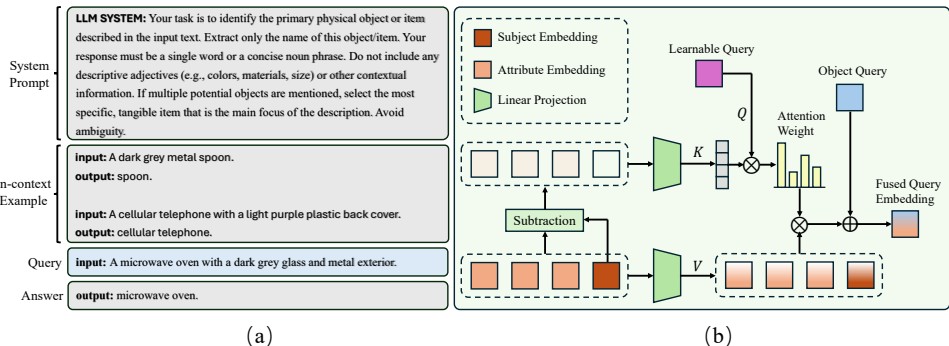

|  |  |
|---|---|
| (a) | (b) |

Figure 3: (a) Illustration of the prompt for subject identification. The prompt for extracting the associated attributes is shown in our supplementary document. (b) The architecture of the attribute-fused attention layer.

## 3.2 Coarse-grained Object Detection

After the identification stage, we perform coarse-grained object detection(CGOD) by a detection transformer. To retain a relevant attribute cues while avoiding over-representation, we also propose an attribute embedding fusion module to exploit the helpful fine-grained attributes in an attention-based manner.

Specifically, we adopt the LAMI-DETR[5] as our baseline transformer, which is built upon the DINO[3] detection framework. Concretely, an input image is encoded to a frozen ConvNext[22] backbone from the pre-trained CLIP image encoder to output the spatial feature map $\mathbf{F}_{\text{conv}}$. The spatial feature map is then input to a learnable transformer encoder for refinement. The refined encoder feature is denoted as $\mathbf{F}_{\text{enc}}$.

**Attribute Embedding Fusion**. After obtaining the refined feature map, we introduce an Attribute Embedding Fusion module to fuse the subject embeddings and attribute embeddings into input object queries of the transformer decoder. Specifically, we follow LaMI-DETR to select the top $k$ pixels in the encoder feature $\mathbf{F}_{\text{enc}}$ as object queries based on their classification logits. This process is formulated as,

$$\{\mathbf{q}_j\}_{j=1}^{K} = \text{Top}_k(\max_i(\text{CLS}(\{\mathbf{t}_i\}_{i=1}^{n}, \mathbf{F}_{\text{enc}}))), \tag{1}$$

Here $\text{CLS}(\{\mathbf{t}_i\}_{i=1}^{n}, \cdot)$ is the classifier which inputs the subject embeddings as frozen layer weights and outputs the logits corresponding to the $n$ classes. With the TopK selection, each selected query is matched with a class with the largest classification scores. Then the query embeddings are fused with the corresponding subject embedding and attribute embeddings through a cross-attention layer, which is formulated as follows,

$$\mathbf{q}_i^{\text{f}} = \mathbf{q}_i + \text{ATTN}(\mathbf{q}^{\text{l}}, \{\mathbf{t}_i - \mathbf{t}_i\} \cup \{\mathbf{t}_i^{j} - \mathbf{t}_i\}_{j=1}^{n_j}, \{\mathbf{t}_i\} \cup \{\mathbf{t}_i^{j}\}_{j=1}^{n_j}). \tag{2}$$

The union of subject embedding $\mathbf{t}_i$ and fine-grained attribute embeddings $\{\mathbf{t}_i^{j}\}_{j=1}^{n_j}$ for the matched class $i$ is used as the value states of the cross-attention layer. We adopt the difference between the attribute embeddings and the corresponding subject embedding as key states to highlight the attribute information. $\mathbf{q}^{\text{l}}$ is a learnable query. The architecture of attribute-fused attention is shown in Figure 3(b). This attention layer dynamically integrates the subject embedding and attribute embeddings into queries, enabling the detector to exploit helpful attributes for detection.

After that, the fused query embeddings $\{\mathbf{q}_i^{\text{f}}\}$ and the encoder feature $\mathbf{F}_{\text{enc}}$ are input to a DINO decoder to output the prediction embedding $\{\mathbf{p}_i\}_{i=1}^{k}$. We adopt the aforementioned classifier $\text{CLS}(\{\mathbf{t}_i\}_{i=1}^{n}, \cdot)$ to output the detector's coarse confidence $\mathbf{s}_j^{\text{coarse}} \in \mathbb{R}^{(n)}$ for the $j$-th prediction with a sigmoid function,

$$\begin{aligned} \mathbf{l}_j^{\text{coarse}} &= m_{\text{coarse}} \text{CLS}(\{\mathbf{t}_i\}_{i=1}^{n}, \mathbf{p}_j), \\ \mathbf{s}_j^{\text{coarse}} &= \text{Sigmoid}(\mathbf{l}_j^{\text{coarse}}), \end{aligned} \tag{3}$$

where $m_{\text{coarse}}$ is a scaling factor. The bounding box locations $\{\mathbf{b}_i\}_{i=1}^{k}$ are output from the prediction embedding with a box regression process.

### 3.3 Fine-grained Attribute Discrimination

The bounding boxes output by the detection transformer are used as candidate regions of fine-grained classes. To discriminate whether the fine-grained attribute is aligned with the candidate bounding boxes, we introduce a fine-grained attribute discrimination(FGAD) module. To this end, we first add a lightweight linear projection layer after the frozen CLIP text encoder to refine the text embeddings for better representation on fine-grained attributes. This lightweight linear projection layer bridges the gap between general-purpose pre-trained representations and task-specific attribute semantics, enhancing discriminative capability for subtle attribute distinctions. Then the full fine-grained class names are fed into the refined CLIP text encoder $T'$ to obtain the corresponding fine-grained class embeddings $\{\hat{\mathbf{t}}_i\}_{i=1}^n$. Subsequently, the image embeddings of each candidate are generated from the spatial feature map $\mathbf{F}_{\text{conv}}$ from the frozen ConvNext backbone by performing a pooling operation on the features of the box location. Finally, we estimate an attribute similarity score for each candidate box, noted as $\mathbf{s}^{\text{fine}}$. The attribute similarity score $\mathbf{s}_j^{\text{fine}} \in \mathbb{R}^{(n)}$ of the $j$-th prediction is estimated from,

$$\mathbf{l}_j^{\text{fine}} = m_{\text{fine}} \cos(\text{Pooling}(\mathbf{F}_{\text{conv}}[\mathbf{b}_j]), \{\hat{\mathbf{t}}_i\}_{i=1}^n),$$
$$\mathbf{s}_j^{\text{fine}} = \text{Softmax}(\mathbf{l}_j^{\text{fine}}). \tag{4}$$

The $\cos(\cdot)$ is the cosine similarity function, and $m_{\text{fine}}$ is a scaling factor. Since the spatial feature map $\mathbf{F}_{\text{conv}}$ is previously calculated in the coarse-grained detection module, the fine-grained attribute discrimination only slightly increases the inference time In addition to CLIP, other VLMs can also be applied to calculate the attribute similarity score, as shown in our experiments. Nevertheless, the application of other VLMs always substantially increases the inference time. The final scores of the $j$-th bounding box prediction for the fine-grained detection are set as,

$$\mathbf{s}_j^{\text{final}} = (\mathbf{s}_j^{\text{coarse}})^{(\alpha)}(\mathbf{s}_j^{\text{fine}})^{(1-\alpha)}, \tag{5}$$

where $\alpha$ is a hyperparameter.

### 3.4 Training Objective

We introduce a two-stage training pipeline for the GUIDED. During training, we first follow the LaMI-DETR[5] training pipeline to pretrain the detection transformer on the base classes of the traditional open vocabulary dataset without applying the attribute similarity score. In this stage, the class names of the base classes are extended by a large language model to construct the attribute embeddings for the attribute embedding fusion module. In the second stage, we introduce the FG-OVD dataset to train the detection transformer and the refined CLIP. The fine-grained class names in the FG-OVD dataset are decomposed into coarse-grained subjects and attributes by the subject identification. For the detection transformer, we generate the ground truth with the labels of subjects to supervise the detection transformer. Furthermore, we introduce an additional binary loss for the samples on the original fine-grained classes,

$$L_{\text{fine}} = -\sum_j \text{gt}_j \log((\mathbf{s}_j^{\text{coarse}})^{(\alpha)}(\mathbf{s}_j^{\text{fine}})^{(1-\alpha)}). \tag{6}$$

Here $\text{gt}_j$ is the one-hot class label of the $j$-th sample. This loss enhances the discriminative capability for attribute distinctions of embeddings from the refined CLIP for improved alignment.

## 4 Experiments

### 4.1 Dataset

Our method is evaluated on the two benchmark fine-grained open vocabulary object detection datasets, FG-OVD and 3FOVD. Additionally, we perform training on the LVIS dataset.

**FG-OVD**. Fine-grained open vocabulary detection (FG-OVD) dataset [2] is an evaluation task for comprehensively evaluating the fine-grained discrimination capabilities of models. Each annotation in FG-OVD is paired with a positive caption and up to ten hard negatives generated by substituting attribute words while preserving sentence structure. The data are partitioned into four difficulty splits (Trivial, Easy, Medium, Hard) and four attribute-focused subsets (Colour, Material,

Pattern, and Transparency). We follow the official benchmarks subset splits and report mean average precision[17](mAP) averaged over all eight tracks.

**3F-OVD**. The recently released 3F-OVD [21] benchmark provides a more demanding test-bed for fine-grained open-vocabulary detection under long-caption queries, which assigns a single, sentence-length description to every class, and re-uses that description across all images that contain the class. The benchmark comprises two distinct domains: vehicles (NEU-171K-C) with 598 fine-grained classes, and retail products (NEU-171K-RP) with 121 fine-grained classes. Consistent with the benchmark authors' configuration, we report mAP across both domains.

**LVIS**. The LVIS dataset is a long-tailed object detection dataset with . Following the open vocabulary setting in LaMI-DETR[5], 866 common and frequent categories in the LVIS dataset are set as base classes, while the remaining 335 rare categories are set as novel classes. We mainly use the base classes in LVIS for training.

## 4.2 Implemental Details

We mainly adopt the LaMI-DETR[5] as our codebase. The ConvNext backbone of the detection transformer is initialized from ConvNeXt-Large-D-320[22] in OpenCLIP[10]. We follow GroundingDino[20] to retain the top-k = 900 tokens ranked by coarse-grained classification logits. In the two-stage training pipeline, we first pre-train on the base-class subset of the LVIS for 85200 iterations; we then fine-tune for a further 2000 iterations on the FG-OVD training set. For 3FOVD, we extract all the available captions' subjects using LLM and process all the classes together. More details are presented in our supplementary document. At the score ensemble process, the $\alpha$ is set to 0.6. $m_{\text{fine}}$ and $m_{\text{coarse}}$ is set to 100. We leave other hyper-parameters the same as in LaMI-DETR.

## 4.3 Experimental Results

**Comparison on FG-OVD dataset.** We compare GUIDED against the existing OVD methods on FG-OVD datasets, including OWL-ViT[24], Detic[41], ViLD[8], Grounding DINO[20], CORA[33], and OV-DINO[29]. Furthermore, we apply GUIDED to three distinct architecture-based OVD models, Grounding DINO, OWL-ViT, and LaMI-DETR[5]. Note that the attribute embedding fusion module is only applied to the DETR-based framework, LaMI-DETR. The results are shown in Table 1. The mAP performance of GUIDED significantly surpasses that of other OVD methods. Specifically, GUIDED achieves a 23.2% mAP improvement over our baseline method LaMI-DETR, highlighting the effectiveness of leveraging PVLMs for fine-grained attribute discrimination in GUIDED. While methods like Grounding DINO and OV-DINO benefit from large-scale pretraining on coarse-grained datasets (Object365[26], GoldG[11]), they exhibit limited capability in distinguishing fine-grained categories. As shown in the table, GUIDED is capable of enhancing the performance of these methods on the FG-OVD. Furthermore, our methods defeat existing FG-OVD methods by a large margin. Specifically, HA-FGOVD[23] only slightly improves the performance since it only modifies the input text embeddings. In contrast, GUIDED boosts the fine-grained detection capability of OVD methods, underscoring the generalization of our methods.

**Comparison on 3FOVD dataset.** We conduct evaluation of our proposed GUIDED on the 3FOVD dataset in Table 2. Compared with FG-OVD, 3FOVD is a more complex task since the class names in 3FOVD are more complicated proper nouns(e.g. Drink_Coca-Cola, Car_Porsche-macan) with corresponding captions. Note that we do not perform training on 3FOVD but only transfer the model trained on LVIS and FG-OVD to conduct the evaluation. In 3FOVD, we extract the superclasses of the fine-grained class names as subjects for coarse-grained object detection and utilize the caption data as the fine-grained full names in fine-grained attribute discrimination. The results show our method defeats other methods by clear margins. Compared with LaMI-DETR, our method demonstrates a clear improvement on both subsets, underscoring the effectiveness of GUIDED on the detection of complicated fine-grained classes.

## 4.4 Ablation Study and Analysis

**Ablation of key factors in the GUIDED framework**. We conducted an ablation study to assess the effectiveness of each key factor in our proposed GUIDED framework in Table 4. For comparison, we train the LaMI-DETR on FG-OVD with different training strategies, including training from

Table 1: mAP evaluation results on FG-OVD benchmark (%). The performance in 'Average' is the average of performance over the 8 sub-datasets. 'Trans.' denotes the performance on the Transparency subset. 'Finetune' denotes finetune the LaMI-DETR on FG-OVD training set. Here * denotes the results are from our reproduction.

| Detector | Hard | Medium | Easy | Trivial | Color | Material | Pattern | Trans. | Average |
|---|---|---|---|---|---|---|---|---|---|
| OWL-ViT(B/16) | 26.2 | 39.8 | 38.4 | 53.9 | 45.3 | 37.3 | 26.6 | 34.1 | 37.7 |
| OWLv2(B/16) | 25.3 | 38.5 | 40.0 | 52.9 | 45.1 | 33.5 | 19.2 | 28.5 | 35.4 |
| OWLv2(L/14) | 25.4 | 41.2 | 42.8 | 63.2 | 53.3 | 36.9 | 23.3 | 12.2 | 37.3 |
| Detic | 11.5 | 18.6 | 18.6 | 69.7 | 21.5 | 38.8 | 30.1 | 24.6 | 29.3 |
| ViLD | 22.1 | 36.1 | 39.9 | 56.6 | 43.2 | 34.9 | 24.5 | 30.1 | 35.9 |
| CORA | 13.8 | 20.0 | 20.4 | 35.1 | 25.0 | 19.3 | 22.0 | 27.9 | 22.9 |
| OV-DINO | 18.6 | 28.4 | 25.0 | 54.3 | 35.6 | 30.0 | 21.0 | 24.2 | 29.6 |
| Grounding DINO | 17.0 | 28.4 | 31.0 | 62.5 | 41.4 | 30.3 | 31.0 | 26.2 | 33.5 |
| + HA-FGOVD | 19.2 | 32.3 | 34.0 | 62.2 | 41.5 | 33.0 | 32.1 | 29.2 | 35.4 (+1.9) |
| + GUIDED | 35.1 | 49.3 | 52.8 | 57.1 | 49.7 | 57.0 | 26.4 | 39.6 | 45.9 (+12.4) |
| OWL-ViT(L/14) | 26.6 | 39.8 | 44.5 | 67.0 | 44.0 | 45.0 | 36.2 | 29.2 | 41.5 |
| + HA-FGOVD | 31.4 | 46.0 | 50.7 | 67.2 | 48.4 | 48.5 | 38.0 | 32.7 | 45.4 (+4.3) |
| + GUIDED | 46.8 | 59.4 | 64.1 | 66.2 | 60.4 | 58.9 | 44.7 | 54.5 | 56.9 (+15.4) |
| LaMI-DETR | 29.2 | 40.6 | 42.9 | 63.5 | 49.5 | 39.2 | 34.6 | 46.2 | 43.2 |
| + Finetune | 39.5 | 50.7 | 54.2 | 66.0 | 51.9 | 53.7 | 42.1 | 49.1 | 50.9 (+7.7) |
| + HA-FGOVD* | 33.5 | 45.9 | 47.5 | 63.8 | 52.7 | 42.6 | 36.9 | 50.1 | 46.6 (+3.4) |
| + GUIDED | **57.5** | **69.5** | **73.3** | **72.6** | **64.8** | **68.5** | **62.0** | **63.4** | **66.4** (+23.2) |

Table 2: mAP evaluation results on the 3FOVD benchmark(%).

| Method | NEU-171K-C | NEU-171K-RP |
|---|---|---|
| Detic | $6.6 \times 10^{-4}$ | $2.2 \times 10^{-2}$ |
| Vild | $3.8 \times 10^{-4}$ | $1.1 \times 10^{-2}$ |
| GroundingDino | $1.3 \times 10^{-3}$ | $7.6 \times 10^{-4}$ |
| LaMI-DETR | $9.0 \times 10^{-4}$ | $2.3 \times 10^{-1}$ |
| + GUIDED | $\mathbf{7.2 \times 10^{-3}}$ | $\mathbf{2.7 \times 10^{-1}}$ |

Table 3: Ablation with different VLMs applied in FGOD. 'Time' denotes the averaged inference time(ms) for each image.

| VLM | mAP | Time |
|---|---|---|
| CLIP (T) | 46.9 | 210.3 |
| LLaVA-1.6 | 51.2 | 34718.0 |
| Refined CLIP (T') | 60.8 | 212.1 |

scratch and fine-tuning. As shown in the table, finetuning LaMI-DETR with FG-OVD performs much better than training from scratch, showing the significance of the first-stage training on LVIS. With GUIDED, the performance improves from 50.9% to 62.4%, underscoring the superiority of GUIDED training strategies. We also tease apart the key modules in GUIDED to conduct the ablations. Integrating the attribute embedding fusion(AEF) in GUIDED leads to a performance gain of 4.0%, validating the capability of AEF to selectively integrate fine-grained to improve the capability of the detector. 'GUIDED w/o CGOD' denotes that we directly adopt the full embeddings of fine-grained classes in the detector to achieve detection of fine-grained classes instead of coarse-grained subjects.Performing coarse-grained object detection improves the mAP by 9.4%, showing the effectiveness of task decomposition idea in GUIDED. When removing the projection layer in fine-grained attribute discrimination, the performance decreases by 2.3%, which validates the importance of the projection layer on text embedding generation to represent the fine-grained attribute.

**Robustness of LLMs on subject identification.** To evaluate the robustness of LLMs on subject identification, we manually annotate the subjects from 300 fine-grained classes and assess the accuracy of subject identification with different LLMs, including LLaMA-3.1-8B[7], LLaMA-3.3-70B and GPT-4o. As summarized in Table 5, failure cases are categorized into two types: (1) Hallucination toward in-context samples, where the LLM generates subjects irrelevant to the input text; (2) Other errors, such as identifying an attribute instead of an object. Overall, all three LLMs achieve high correctness rates, demonstrating robust performance across diverse architectures. The results confirm the high robustness of this stage. Crucially, they show that while the smaller LLaMA-3.1-8B model is prone to hallucination errors that always propagate (8/8), this critical failure mode is completely eliminated by larger open-source models. Both LLaMA-3.3-70B and GPT-4o exhibit near-perfect

Table 4: The ablation of key factors in the GUIDED framework on the FG-OVD dataset. The results are shown in mAP (%). 'Baseline' denotes the LaMI-DETR. 'AEF' denotes the attribute embedding fusion module. 'CGOD' denotes the coarse-grained object detection.

| Method | Hard | Medium | Easy | Trivial | Color | Material | Pattern | Transp. | Average |
|---|---|---|---|---|---|---|---|---|---|
| Baseline | 29.2 | 40.6 | 42.9 | 63.5 | 49.5 | 39.2 | 34.6 | 46.2 | 43.2 |
| + Train from scratch | 31.7 | 39.6 | 43.4 | 42.6 | 32.4 | 33.0 | 30.8 | 20.4 | 34.2 |
| + Finetune | 39.5 | 50.7 | 54.2 | 66.0 | 51.9 | 53.7 | 42.1 | 49.1 | 50.9 |
| + GUIDED w/o AEF | 53.0 | 65.2 | 69.5 | 72.0 | 64.1 | 63.3 | 49.0 | 63.4 | 62.4 |
| + GUIDED w/o CGOD | 48.6 | 59.7 | 64.2 | 64.3 | 58.9 | 57.8 | 45.6 | 56.6 | 57.0 |
| + GUIDED w/o projection | 57.5 | 67.8 | 70.2 | 70.8 | 64.0 | 66.3 | 55.4 | 60.8 | 64.1 |
| + GUIDED | **57.5** | **69.5** | **73.3** | **72.6** | **64.8** | **68.5** | **62.0** | 63.4 | **66.4** |

performance, with their rare errors being minor and not always affecting the final detection. This demonstrates that our method's success is not tied to a specific proprietary model and is robust when using state-of-the-art open-source alternatives.

Furthermore, we also report the detection performance with open-source LLMs as alternatives to GPT-4o. As quantified in Table 6, the mAP of GUIDED drops by merely 0.5% with LLaMA-3.1-8B and improves by 0.1% with LLaMA-3.3-70B. This demonstrates that GUIDED performance is not dependent on a specific proprietary model. We will provide more results with LLaMA-3.1-8B and LLaMA-3.3-70B for other OVD models and other datasets in our revised paper for reproducibility.

Table 5: The number of failure cases in subject detection of 300 samples in the FG-OVD dataset with different LLMs. The notation '$x/y$' denotes that there are $y$ failure cases, of which $x$ lead to detection errors.

| LLM | Hallucination | Others | Total |
|---|---|---|---|
| GPT4-o | 0/0 | 1/2 | 1/2 |
| LLaMA-3.1-8B | 8/8 | 1/1 | 9/9 |
| LLaMA-3.3-70B | 0/0 | 1/1 | 1/1 |

Table 6: The comparison of mAP(%) evaluation results on the FG-OVD benchmark with different LLMs in subject identification.

| LLM | mAP |
|---|---|
| GPT4-o | 66.4 |
| LLaMA-3.1-8B | 65.9 |
| LLaMA-3.3-70B | 66.5 |

**Analysis of text embeddings applied in CGOD and FGAD.** We also conduct an ablation study about the text embeddings applied in CGOD and FGAD, which is illustrated in the table 7. Applying the refined text encoder T' with the lightweight projection layer instead of the original CLIP text encoder T improves the mAP by 2.3% in FGAD but leads to a performance degradation of 5.8% in CGOD. This shows that the refined text encoder enhances the fine-grained discrimination capability in FGAD while exacerbating overfitting on the base classes in CGOD. GUIDED only use the fine-grained text encoder in FGAD, achieving an optimal solution. Furthermore, we observe that using the full names of the fine-grained classes in object detectors results in a significant performance drop, validating the necessity of task decomposition for the FG-OVD task.

Table 7: The ablation of generated text embeddings of fine-grained classes. 'TE' denotes the text encoder used for embedding generation. 'Coarse' and 'Full' denote the coarse-grained subject and full fine-grained class name used for generation, respectively.

| CGOD | | FGAD | | mAP |
|---|---|---|---|---|
| TE | Text | TE | Text | |
| T | Full | T | Full | 53.5 |
| T' | Full | T' | Full | 49.8 |
| T | Full | T' | Full | 57.0 |
| T | Coarse | T | Full | 64.1 |
| T' | Coarse | T' | Full | 60.3 |
| T | Coarse | T' | Full | 66.4 |

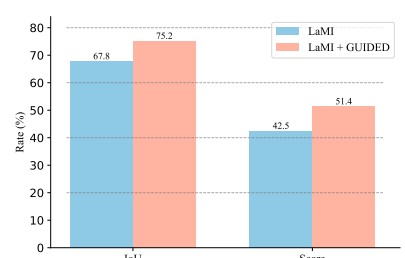

Table 8: The mean classification scores and the mean IoU of the prediction box with the ground truth box of the LaMI-DETR with and without GUIDED.

**More ablations on FGAD.** Our FGAD can be easily integrated with existing PVLMs with different structures. Specifically, we apply LLaVA-1.6(llava-v1.6-mistral-7b)[18] without training to estimate the attribute similarity score by prompting LLaVA with "Does this image match the attributes described in the following caption? If so, output yes, if not, output no" on the coarse-grained box regions of images. The attribute similarity scores are obtained from the probability of generating "yes" tokens. For fair comparison, we directly apply different PVLMs in FGAD with the detector after the first stage of training on LVIS dataset. As presented in Table 3, LLaVA achieves a higher score than the pretrained CLIP but lower scores than the refined CLIP, showing the generalization of our GUIDED framework in integrating different PVLMs in FGAD. Although LLaVA achieves encouraging performance, the inference speed of LLaVA is much lower than that of CLIP.

**Inference time analysis.** To provide an assessment of the computational overhead in GUIDED, we report the inference time of detecting one class in an image using locally deployable LLMs: LLaMA-3.1-8B and LLaMA-3.3-70B[7]. As shown in the Table 9, the inference time increase in GUIDED primarily stems from LLM-based subject identification, while AEF and CLIP design in FGAD contribute minimally to latency. This represents a trade-off between our method's enhanced semantic understanding and computational cost. Nevertheless, the LLM-based subject identification is performed once per class name, not per image. For any given dataset or application scenario, the set of fine-grained classes is fixed. Therefore, the parsing results can be pre-computed and cached offline, imposing no additional LLM-related latency. For scenarios requiring on-the-fly parsing of new class names, the latency can indeed be a factor. This can be alleviated by employing more lightweight LLMs or batch processing multiple subject identification tasks in one chat.

Table 9: The comparison of inference time(ms) between Baseline(LaMI-DETR) and GUIDED for detecting one class in an image. We also report the average mAP(%) of each method in FGOVD.

| Method | LLM | Detector | CLIP | Overall | mAP |
|---|---|---|---|---|---|
| Baseline | - | 198.5 | 13.2 | 211.7 | 43.2 |
| GUIDED with LLaMA-3.1-8B | 72.1 | 198.8 | 13.3 | 284.2 | 65.9 |
| GUIDED with LLaMA-3.3-70B | 193.8 | 198.8 | 13.3 | 405.9 | 66.5 |

**More analysis of GUIDED.** Furthermore, we present the mean classification scores and the mean IoU of the prediction box with the ground truth box of the LaMI-DETR with and without GUIDED in Figure 8. The results reveal that our GUIDED enhances both the capability of localization and confidence with the subject embedding and the attribute embedding fuse module, demonstrating the superiority of our method.

## 5 Limitations and Conclusions

**Limitations.** While GUIDED achieves strong performance on isolated fine-grained object recognition, the attribute discrimination operates on features within coarse-level detection boxes. When relevant attributes extend beyond the localized regions, performance may degrade. This could be mitigated by using expanded region proposals or incorporating context-aware reasoning beyond bounding boxes.

**Conclusions.** In this work, we present GUIDED, a decomposition framework for fine-grained open-vocabulary object detection. By explicitly decoupling object localization and fine-grained attribute discrimination, GUIDED addresses the core challenge of semantic entanglement in vision-language embeddings. Through task-specific modeling and selective attribute integration, our approach leverages the strengths of both detection transformers and pretrained vision-language models. Extensive experiments demonstrate that GUIDED achieves state-of-the-art performance across multiple FG-OVD benchmarks, highlighting the effectiveness of task decomposition for fine-grained visual understanding under open-vocabulary settings.

## 6 Acknowledgments

This work is supported in part by the National Key R&D Program of China (2024YFB3908503, 2024YFB3908500), in part by the National Natural Science Foundation of China (62322608), in part by the Shenzhen Science and Technology Program (NO. JCYJ20220530141211024) and in part by the Guangdong Basic and Applied Basic Research Foundation under Grant 2024A1515010255.

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
