# OpenReview forum: "GUIDED: Granular Understanding via Identification, Detection, and Discrimination for Fine-Grained Open-Vocabulary Object Detection"
_NeurIPS.cc/2025/Conference — NeurIPS 2025 poster_

### Official Review · Reviewer_HXfJ · 2025-06-17

**Clarity:** 3
**Significance:** 4
**Originality:** 3
**Rating:** 4
**Confidence:** 4

**Summary:**

The paper addresses the challenge of fine-grained open-vocabulary object detection (FG-OVD), where existing methods struggle with semantic entanglement between object subjects and attributes in vision-language model (VLM) embeddings. The proposed GUIDED framework decomposes FG-OVD into three stages: subject identification using a large language model (LLM), coarse-grained object detection guided by subject embeddings, and fine-grained attribute discrimination via a refined VLM. By separating localization and attribute recognition, GUIDED mitigates attribute over-representation and semantic drift. Key innovations include an attribute embedding fusion module for selective attribute integration and a projection-based discrimination mechanism. Experiments on FG-OVD and 3F-OVD benchmarks show GUIDED outperforms state-of-the-art methods by up to 19.8% in mAP, demonstrating improved fine-grained detection capability.

**Questions:**

1. It is not aligned with Figure 3(b) and Figure 2. For example, in Figure 3(b), there are two kinds of queries: a learnable query and an object query.
2. What the mean of EDD in Tables 1 and 2?

**Ethical Concerns:**

["NO or VERY MINOR ethics concerns only"]

**Final Justification:**

Thanks for the response from the authors. My questions have been addressed; The insights of this work are novel to me, and the experimental improvements are impressive. However, this method also introduces additional LLM computations; therefore, I keep my rating unchanged.

**Limitations:**

The authors have discussed the limitations.

**Quality:**

3

**Strengths And Weaknesses:**

### Strengths
1. The attribute embedding fusion and projection-based discrimination modules are plug-and-play, enhancing existing OVD frameworks without requiring major architectural changes.
2. GUIDED effectively decouples FG-OVD into manageable subtasks, leveraging the strengths of LLMs for subject parsing, detection transformers for localization, and VLMs for attribute discrimination. This addresses the core issue of semantic entanglement.
3. Experiments cover multiple benchmarks (FG-OVD, 3F-OVD), architecture variants (Grounding DINO, OWL-ViT, LaMI-DETR), and ablation studies validating each module's contribution.

### Weaknesses
1. The motivation is in contrast to previous methods that argue that the attributes will help the open-vocabulary object detection, such as [1].
2. During inference, the proposed method needs an LLM to infer the subject name; it would significantly decrease the inference efficiency compared with previous OVD methods, i.e., the embedding for class names is precomputed for inference.
3. Due to the usage of LLM, the proposed method also needs to be compared with the results of directly using the MLLM for FG-OVD.

[1] OvarNet: Towards Open-vocabulary Object Attribute Recognition.

---

> ### Author Rebuttal · Authors · 2025-07-31
>
> Thanks for the positive feedback on the effectiveness of modules in our method.  We also appreciate the constructive comments to improve the paper. We would like to answer the proposed questions in the following:
>
> **The motivation is in contrast to previous methods.**
> Thank you for raising this important point. Our motivation does not contradict, but rather builds upon and extends the principles of methods like OvarNet to address the unique challenges of FG-OVD.
> First, we concur with OvarNet's core idea that learning attributes of objects can be beneficial for object detection. Our AEF module is designed on this principle, aiming to fuse beneficial attribute information with object features. In this regard, our philosophy is aligned with prior work.
> Second,  unlike conventional OVD (e.g., on MS-COCO) or attribute detection in OvarNet, FG-OVD involves class names that are dense with descriptive attributes (e.g., "blue striped shirt"). This co-occurrence can lead to an "attribute over-representation" problem in text embeddings, where attribute semantics overwhelm the core subject identity. OvarNet, while effective for its target tasks, does not explicitly address this semantic entanglement issue inherent to FG-OVD. In summary, our work proposes to leverage helpful attributes while mitigating their potential to obscure the subject's identity in the fine-grained text embeddings. We will refine our manuscript to make this distinction clearer.
>
> **Inference efficiency decreased by LLM.**
> We acknowledge the reviewer's valid concern regarding the inference latency introduced by the LLM. This represents a trade-off between our method's enhanced semantic understanding and computational cost. Nevertheless, the LLM-based subject identification is performed once per class name, not per image. For any given dataset or application scenario, the set of fine-grained classes is fixed. Therefore, the parsing results can be pre-computed and cached offline, imposing no additional LLM-related latency.
> For scenarios requiring on-the-fly parsing of new class names, the latency can indeed be a factor. This can be alleviated by employing more lightweight LLMs(e.g. LLaMA-3.1-8B as shown in Table 1) or batch processing multiple subject identification tasks in one chat with LLMs. We will also provide the inference efficiency analysis in our main paper.
>
> **Table 1:** The comparison of inference time (ms) between Baseline (LaMI-DETR) and GUIDED for detecting one class in an image. We also report the average mAP (%) of each method in FGOVD.
>
> | Method                        | LLM   | Detector | CLIP  | Overall | mAP  |
> |-------------------------------|-------|----------|-------|---------|------|
> | Baseline                      | -     | 198.5    | 13.2  | 211.7   | 43.2 |
> | GUIDED with LLaMA-3.1-8B      | 72.1  | 198.8    | 13.3  | 284.2   | 65.9 |
> | GUIDED with LLaMA-3.3-70B     | 193.8 | 198.8    | 13.3  | 405.9   | 66.5 |
>
> **Performance of MLLMs.**
> We evaluate a `LLaVA-v1.6-Mistral-7B`, one of the strongest open-source MLLMs available.
> For each class in an FG-OVD annotation, we queried the model with a RefCOCO-style prompt, where `statement` is replaced by the target class name:
>
> > `Your task is to locate and identify all objects or regions in the image that match the provided textual description.`
> > `Description: "{statement}"`
> > `Please return a JSON list. Each element must contain a "bbox" in the form [x1, y1, x2, y2] and a "confidence" between 0.0 and 1.0.`
>
> Under this protocol, the model attains only **0.4~mAP** on the FG-OVD benchmark, which is significantly below our *GUIDED*. This experiment reveals that direct MLLM inference is presently insufficient for practical or research use in FG-OVD.
>
> **It is not aligned with Figure 3(b) and Figure 2.**
> Thanks for pointing this out. Figure 2 does not accurately depict the two query types. We will correct it in our revised paper.
>
> **The meaning of 'EDD'.**
> We apologize for the oversight. 'EDD' was a working name for our method from an earlier draft. We will correct this typo in the revised manuscript and perform a thorough check to ensure all terminology is consistent.

---

### Official Review · Reviewer_vfih · 2025-06-28

**Clarity:** 1
**Significance:** 2
**Originality:** 3
**Rating:** 4
**Confidence:** 4

**Summary:**

This paper introduces GUIDED, a method for fine-grained open-vocabulary object detection that addresses the challenge of semantic entanglement between object categories and their attributes. The proposed approach decomposes the task into three stages:

1. Subject Identification, where GPT-4o is used to extract a coarse object category and its associated attributes from fine-grained class names.

2. Coarse-Grained Object Detection, which uses only the subject embedding for object localization, with an attribute embedding fusion module to selectively integrate relevant attribute cues.

3. Fine-Grained Attribute Discrimination, which re-scores detected regions using full fine-grained descriptions via refined CLIP text encoder.

The model is trained in two stages, first on LVIS base categories, then fine-tuned on FG-OVD. The model is evaluated on the FG-OVD and 3F-OVD benchmarks.

**Questions:**

In the attribute embedding fusion module, query updates are computed using cross-attention where the keys are formed by subtracting the subject embedding from each attribute embedding. While the idea is that this highlights the attribute-specific signal, it’s not clear to me:

- Why this subtraction should be meaningful or stable, especially across diverse classes or longer prompts. For example, subtracting the embedding of “car” from “a shiny red car with tinted windows” assumes that the resulting vector cleanly captures the difference (“shiny red with tinted windows”). But CLIP embeddings are not trained with this kind of linear structure in mind, and it's unclear whether such a subtraction generalizes across different sentence structures or domains.

- Whether it introduces noise or ambiguity when attributes are semantically similar or visually correlated. For example, the difference between “a white ceramic mug” and “a white porcelain mug” may be minimal in CLIP space and subtracting the subject (“mug”) from each might produce almost the same or unstable vectors. This could affect attention-based fusion especially when the attributes are subtle or highly co-occurring.

Has this subtraction-based attention been compared empirically with simpler alternatives like concatenation or additive fusion?

Clarifying this design choice and providing results on standard open-vocabulary settings like COCO and LVIS (especially given that the model is trained on LVIS base) would help in assessing generalization and could positively influence my rating.

**Ethical Concerns:**

["NO or VERY MINOR ethics concerns only"]

**Final Justification:**

After reading the author response and the other reviewers’ comments, I am updating my rating to a borderline accept. The paper presents a meaningful and novel decomposition strategy for fine-grained open-vocabulary object detection (FG-OVD), separating subject localization and attribute discrimination into distinct modules. The attribute embedding fusion (AEF) module is conceptually intuitive and empirically validated through targeted ablations. The method shows good performance on FG-OVD benchmarks and modest results on general OVD tasks like LVIS and COCO.

While I appreciate the authors' detailed clarifications and additional results, I remain concerned about the way some evaluation tables mix results from different training stages (e.g., FG-OVD after Stage 2 and LVIS after Stage 1), which could convey misleading conclusions about generalization. A clearer separation or consistent reporting of results from the same model would improve confidence in the claims. Nonetheless, I acknowledge the effort made to address concerns and am willing to lean towards acceptance.

**Limitations:**

Yes

**Paper Formatting Concerns:**

No major formatting issues.

**Quality:**

2

**Strengths And Weaknesses:**

The following are the strengths of the paper.

1. The paper provides a reasonable rationale for separating fine-grained object detection into three stages, subject identification, coarse-grained object detection and fine-grained attribute discrimination. The observation that VLMs like CLIP perform better on base-level category names than on attribute-rich prompts is well-argued and aligns with known limitations of CLIP embeddings.

2. The proposed attribute embedding fusion (AEF) module, which integrates attribute information into object queries via cross-attention, is conceptually simple but intuitive. It enables selective incorporation of useful attributes without much interfering with the subject signal during detection.


The following are the weaknesses of the paper.

1. Key variables such as F_conv, F_en, etc. are not linked to the architecture block diagrams. The block diagram fails to capture the structure described in the equations, making it difficult to follow the implementation details.

2. **Unfair comparison with LaMI-DETR:** The paper claims a +23.2% improvement over LaMI-DETR but does not clearly state whether LaMI-DETR was fine-tuned on FG-OVD or whether the same training protocol and dataset were used. Without matched experimental conditions, such comparisons are misleading.

3. **No evaluation on standard open-vocabulary benchmarks:** Despite training on LVIS base categories, the paper skips any evaluation on LVIS novel categories (e.g. APr) or COCO base/novel splits. Even though the focus is fine-grained OVD, it is still important to report performance on broader OVD tasks to understand generalizability and potential trade-offs.

---

> ### Author Rebuttal · Authors · 2025-07-31
>
> Thanks for the positive feedback on the reasonable rationale and the AEF module in our method.  We also appreciate the constructive comments to improve the paper. We would like to answer the proposed questions in the following:
>
> **Key variables are not linked to the architecture block diagrams.**
> We appreciate this observation. In the revised manuscript, we will explicitly annotate Figure 2 to map key variables to their corresponding components in the architecture diagram.
>
> **Unfair comparison with LaMI-DETR**
> To address this concern, we clarify that in Table 1 of our main paper, we initially reported LaMI-DETR's performance trained on LVIS following the HA-FGOVD protocol, which matches the first-stage training in our method. Additionally, we report the performance of LaMI-DETR under the same training protocol in Table 4(see line 3 Baseline + Finetune) in our main paper. Under the same protocol, our GUIDED framework shows a 15.5\% improvement, underscoring the effectiveness of our method. For clarity, we will also integrate the results trained on the same protocol into Table 1 in our main paper.
>
>
> **No evaluation on standard open-vocabulary benchmark.**
> We present the performance on the LVIS benchmark in Table 1. The results show fine-tuning on the FG-OVD dataset degrades the performance on the LVIS benchmark. However, co-training on LVIS and FG-OVD during the second stage not only alleviates this issue but surpasses the original LaMI-DETR baseline while mainly preserving FG-OVD performance.  This enhancement stems from our AEF module's attribute integration and expanded training data from FG-OVD. We will supplement our paper with these analyses.
>
>
> **Table 1:** The comparison on the standard open-vocabulary benchmark LVIS dataset and the FGOVD dataset.
> | Finetuning Data | FG-OVD Average | LVIS $AP_r$ | LVIS $AP_c$ | LVIS $AP_f$ | LVIS $AP$ |
> |---|---|---|---|---|---|
> | Baseline | 62.5 | 43.2 | 39.3 | 43.5 | 41.6 |
> | FG-OVD | 66.4 | 39.3 | 32.6 | 38.7 | 36.2 |
> | FG-OVD & LVIS | 65.5 | 44.4 | 39.4 | 43.7 | 41.9 |
>
>
> **Whether subtraction should be meaningful or stable across diverse classes.**
> The CLIP applies the contrastive learning loss to align the embeddings of images and their corresponding caption. The alignment of texts with fine-grained attributes enables CLIP to encode fine-grained attributes in the text embeddings. Nevertheless, we acknowledge that CLIP text embeddings may not always capture all the attributes. To address this issue, GUIDED strategically employs subtraction-based AEF  to selectively exploit the knowledge in attributes for coarse-grained detection, which detects candidate regions for subjects of fine-grained classes \textbf{without requiring exhaustive attribute knowledge}. GUIDED leaves the discrimination of fine-grained attributes in FGAD. FGAD applies a refined CLIP with an additional linear projection trained on fine-grained data, which demonstrates superior fine-grained feature capture capability.
> Empirically, Table 1 of our supplementary material shows that the subtraction process enhances the MAP by 1.3\%,  demonstrating that applying the subtraction is more reasonable than directly applying the original attribute embeddings. Furthermore, employing the CLIP embeddings of the isolated attributes (e.g., "shiny red with tinted windows") declines the performance by 2.8\%. It is because the embeddings of the isolated attributes do not capture the relation between subjects and attributes.
>
>
>
> **Whether subtraction introduces noise or ambiguity.**
> While CLIP text embeddings may not capture all attributes perfectly, the subtraction process may output similar key embeddings for classes with similar or visually correlated attributes. Nevertheless, similar to the response for the last question, GUIDED inherently mitigates this issue by design. Specifically, AEF selectively integrates attribute knowledge for coarse-grained detection rather than exhaustive attribute discrimination, thus avoiding reliance on subtle inter-attribute distinctions.
> Crucially, when subtraction produces similar key embeddings, their corresponding value embeddings, which are directly obtained from attribute embeddings, exhibit analogous similarity. Consequently, AEF’s selective fusion integrates minimally divergent attribute signals into object queries, which naturally down-weighting noise or irrelevant information.
>
>
>
>
> **Comparing subtraction-based attention with simpler architecture.**
> We compare the mAP of subtraction-based attention with additive fusion and concatenation in FGOVD in Table 2.
> For concatenation, we apply a linear projection layer to transform the dimensions back to the original dimensions. The results show that our AEF outperforms both additive fusion and concatenation by a clear margin, demonstrating the effectiveness of the subtraction-based attention on attribute integrations.
>
> **Table 2:** The comparison of mAP(%) evaluation results on the FG-OVD benchmark between our proposed AEF and simpler architecture.
> | Arch | mAP |
> |---|---|
> | Concentration | 61.1 |
> | Addition | 63.2 |
> | AEF | 66.5 |

---

> > ### Comment · Reviewer_vfih · 2025-08-02
> > **Reviewer Response to Author Rebuttal**
> >
> > Thanks for the rebuttal and the clarifications.
> >
> > That said, I noticed a few inconsistencies in the reported numbers which I would like to clarify:
> >
> > 1. Regarding the LVIS evaluation presented in the rebuttal and the ablation results in the supplementary material (specifically Table 1), there appears to be a mismatch. For instance, the FG-OVD number (66.4) is consistent, but the LVIS numbers (APr, APc, APf, AP) reported in the supplementary material do not match any row in the rebuttal. It would be helpful to clarify what the exact training configurations are for each row in the rebuttal table and the supplementary Table 1 (last row). Please specify the training data used, whether co-training was applied or not, the baseline model, and the evaluation protocol.
> >
> > 2. In the Rebuttal Table 2 (comparing AEF with concatenation and addition), the FG-OVD result for AEF (66.5) does not match the main paper (66.4) or the rebuttal Table 1 (65.5). Also, in the rebuttal text, a 2.8% improvement is mentioned for AEF, but based on Table 1 in the supplementary material, the margin appears to be smaller. Please correct or clarify if I am missing something.
> >
> > 3. For completeness, it would be helpful to include the LVIS results as well for the AEF, addition, and concatenation variants reported in Rebuttal Table 2.
> >
> > 4. Finally, I would still encourage reporting results on COCO or providing a clear justification for not including them in the evaluation.
> >
> > These discrepancies in the reported results introduced some doubts about the correctness and consistency. A `brief but clear` clarification on these may help to reinforce confidence in the work.

---

> > > ### Author Response · Authors · 2025-08-03
> > >
> > > We sincerely thank the reviewer for meticulous reading and for identifying these critical inconsistencies. We apologize for the confusion this has caused. We have carefully reviewed all results and provide a full, unified clarification below.
> > >
> > > **Inconsistency of the LVIS evaluation**
> > > The core of the confusion stems from our reporting results from different stages of our training pipeline to evaluate different aspects of performance. GUIDED applies a two-stage training pipeline, which pretrains the model on LVIS and subsequently finetunes the model on FG-OVD.
> > >
> > > (1) For the LVIS evaluation in supplementary Table 1, we report the performance only with the first stage training on LVIS to show a fair comparison on the standard LVIS OVD task.
> > >
> > > (2 )In the previous Rebuttal Table 1, we report the LVIS performance with the full two-stage training to show generalization and potential trade-offs of models. 'Baseline' indicates the results of the official LaMI-DETR checkpoint. The "62.5" for the Baseline on FG-OVD was a typo, and we have corrected it in the following table.
> > >
> > > We have now created a single, comprehensive table that clarifies the exact configuration for each result.
> > >
> > > **Table 1: The comparison on the standard open-vocabulary benchmark LVIS dataset and the FGOVD dataset.**
> > >
> > > | Method | Stage1 | Stage2 | FG-OVD Average | LVIS $AP_{r}$ | LVIS $AP_{c}$ | LVIS $AP_{f}$ | LVIS AP |
> > > | :--- | :--- | :--- | :--- | :--- | :--- | :--- | :--- |
> > > | Baseline | LVIS | - | 43.2 | 43.2 | 39.3 | 43.5 | 41.6 |
> > > | GUIDED | LVIS | - | 46.8 | 43.9 | 39.2 | 43.7 | 41.8 |
> > > | GUIDED | LVIS | FG-OVD | 66.4 | 39.3 | 32.6 | 38.7 | 36.2 |
> > > | GUIDED | LVIS | FG-OVD & LVIS | 65.5 | 44.4 | 39.4 | 43.7 | 41.9 |
> > >
> > > **Inconsistency in Rebuttal Table 2 and Rebuttal Text**
> > > We apologize for the reporting error.
> > > (1) The correct mAP with the AEF module is 66.4. The "66.5" was a typo in our previous response.
> > >
> > > (2) In the rebuttal text, 2.8% improvement mentioned in the rebuttal arises from a variant of AEF which replaces the key embeddings (e.g., subtraction between “a shiny red car” and "a car") by the CLIP embeddings of the isolated attributes (e.g., "shiny red"). We failed to include the table in our last response. We will add it to the supplementary material.
> > >
> > > **Table 2: The ablation of attribute embedding fusion on the FG-OVD and the LVIS dataset.**
> > >
> > > | Method | FG-OVD Average | LVIS $AP_{r}$ | LVIS $AP_{c}$ | LVIS $AP_{f}$ | LVIS AP |
> > > | :--- | :--- | :--- | :--- | :--- | :--- |
> > > | GUIDED with Isolated attributes | 63.6 | 43.4 | 38.9 | 43.3 | 41.4 |
> > > | GUIDED | 66.4 | 43.9 | 39.2 | 43.7 | 41.8 |
> > >
> > > **More LVIS results.**
> > > Thanks for the suggestion. We include the LVIS results as well for the AEF, addition, and concatenation variants reported in the following table.
> > >
> > > **Table 3: The comparison of evaluation results on the FG-OVD and LVIS benchmark between our proposed AEF and simpler architectures.**
> > >
> > > | Method | FG-OVD Average | LVIS $AP_{r}$ | LVIS $AP_{c}$ | LVIS $AP_{f}$ | LVIS AP |
> > > | :--- | :--- | :--- | :--- | :--- | :--- |
> > > | Concatenation | 61.1 | 41.6 | 37.0 | 41.6 | 39.6 |
> > > | Addition | 63.2 | 43.3 | 39.0 | 43.5 | 41.5 |
> > > | AEF | 66.4 | 43.9 | 39.2 | 43.7 | 41.8 |
> > >
> > > **MS-COCO evaluation**. For completeness, we report the MS-COCO result after our two-stage training pipeline on all MS-COCO 80 classes in the following table.
> > >
> > > **Table 4: The comparison on the MS-COCO dataset.**
> > >
> > > | Method | $AP$ | $AP_{50}$ | $AP_{75}$ |
> > > | :--- | :--- | :--- | :--- |
> > > | Baseline | 41.2 | 55.6 | 45.2 |
> > > | GUIDED | 41.6 | 55.9 | 45.8 |
> > >
> > > We hope these detailed clarifications, unified tables, and additional results have resolved all ambiguities. We are grateful for the opportunity to improve our paper based on this valuable feedback.

---

### Official Review · Reviewer_KZWx · 2025-07-02

**Clarity:** 4
**Significance:** 3
**Originality:** 2
**Rating:** 4
**Confidence:** 4

**Summary:**

The paper proposes GUIDED, a decomposition framework for fine-grained open-vocabulary object detection (FG-OVD) that addresses semantic entanglement between subjects and attributes in vision-language model embeddings. The method decomposes FG-OVD into three stages: (1) subject identification using LLMs to extract coarse-grained subjects and attributes, (2) coarse-grained object detection guided by subject embeddings with an attribute embedding fusion module, and (3) fine-grained attribute discrimination using refined CLIP embeddings with a projection head. GUIDED achieves 23.2% mAP improvement over baseline LaMI-DETR on FG-OVD benchmark and demonstrates state-of-the-art performance across multiple datasets.

**Questions:**

1. How robust is the LLM-based subject identification? What happens when the LLM fails to correctly identify subjects or attributes? Have you analyzed error propagation from this stage?
2. Hyperparameters (m_fine=50, m_coarse=50) appear to be fixed values without sensitivity analysis or justification for these specific choices.
3. Given the reproducibility concerns with using proprietary GPT-4o, have you experimented with open-source alternatives like LLaMA or other accessible models? How does the choice of LLM affect the subject identification quality and overall performance?

**Ethical Concerns:**

["NO or VERY MINOR ethics concerns only"]

**Final Justification:**

The authors have addressed my concerns. I hope the authors include the additional experiments in their manuscript to better support their motivation.

**Limitations:**

yes

**Paper Formatting Concerns:**

No significant formatting issues observed.

**Quality:**

3

**Strengths And Weaknesses:**

Strengths:
1. Well-motivated: The paper clearly identifies and visualizes the semantic entanglement issue in existing FG-OVD methods through empirical analysis (Figure 1), showing how attribute over-representation leads to mislocation and semantic drift.
2. Systematic approach: The three-stage pipeline (identification, detection, discrimination) is well-designed and aligns each subtask with appropriate model strengths - using detection transformers for localization and VLMs for fine-grained discrimination.
3. Comprehensive experiment: Extensive experiments on FG-OVD and 3F-OVD benchmarks. The method shows consistent improvements across different base detectors. Achieves significant performance gains and establishes new state-of-the-art results on multiple benchmarks.

Weaknesses:
1. There's insufficient analysis of the attribute similarity score. Could you provide the result, if alpha in Eq. (5) is 0 or 1? Furthermore, have you tried other fusion strategies, e.g., weighted average?
2. The method introduces additional complexity through LLM preprocessing, refined CLIP inference, and multi-stage processing, but computational cost analysis is insufficient. What is the complete computational overhead breakdown for GUIDED compared to direct fine-grained detection? Please include LLM preprocessing time, additional CLIP inference, and memory requirements.
3. Failure cases: The paper briefly mentions limitations regarding attributes extending beyond detection boxes, but doesn't provide visualization analysis or scenarios where the decomposition strategy might not work well.
4. Reproducibility concerns: The method relies on GPT-4o for subject identification, which is a closed-source model. This creates reproducibility issues and introduces potential non-transparent biases in the decomposition process that cannot be fully analyzed or controlled.

---

> ### Author Rebuttal · Authors · 2025-07-30
>
> Thanks for the positive feedback on the well motivation, systematic approach and systematic approach in our method.  We also appreciate the constructive comments to improve the paper. We would like to answer the proposed questions in the following:
>
> **Insufficient analysis of the attribute similarity score.**  We conducted an ablation study by setting $\alpha$ to extreme values  (summarized in Table 1).
> Setting $\alpha$ to $0$  means relying solely on the detector's coarse confidence scores to estimate final scores and $ L_{\text{contrast}}$, while setting $\alpha$ to $1$ means relying only on the attribute similarity scores.
> As shown, both extremes lead to a significant performance drop. This empirically confirms our hypothesis that a balanced fusion is crucial.
> Regarding alternative fusion strategies, we evaluate the weighted average in Table 2. It shows that weighted multiplication achieves a better performance. Compared with the weighted average, the multiplication enforces a high score on both the coarse confidence score and the attribute similarity simultaneously. This prevents candidates with a poor attribute similarity score from being promoted simply because their confidence scores are high (and vice versa).
>
>
> **Table 1:** More ablation study on the hyperparameter $\alpha$.
> | $\alpha$ | 0.0 | 0.6 | 1.0 |
> |---|---|---|---|
> | MAP | 17.1 | 66.4 | 28.0 |
>
>
> **Table 2:** Ablation study with different fusion strategies.
> | Fusion | MAP |
> |---|---|
> | Weighted Average | 63.1 |
> | Ours | 66.4 |
>
>
> **Insufficient computational overhead analysis.**
> While our method utilizes GPT-4o for its reasoning capabilities, measuring its API inference time directly is problematic as network latency becomes a confounding factor. To provide a meaningful assessment of the computational overhead introduced by our LLM component, we instead evaluated inference times using locally deployable LLMs:` LLaMA-3.1-8B` and `LLaMA-3.3-70B`. Crucially, these models achieve comparable mAP performance to GPT-4o on the FG-OVD task (as validated in our experiments), making them valid proxies for analyzing inference speed. We measured the inference time in Table 3. As shown in the Table, the inference time increase in GUIDED primarily stems from LLM-based subject identification, while AEF and CLIP design in FGAD contribute minimally to latency. This time consumption gain can be alleviated by adopting a lightweight LLM or batch processing multiple subject identification tasks in one chat. For memory consumption in Table 4, the memory footprint of our core detection modules (AEF, FGAD) is negligible. The significant memory increase comes from loading the LLM. We will include these analyses in the revised paper.
>
> **Table 3:** The comparison of inference time(ms) between Baseline(LaMI-DETR) and GUIDED for detecting one class in an image. We also report the average mAP(%) of each method in FGOVD.
> | **Method** | LLM | Detector | CLIP | Overall | mAP |
> |---|---|---|---|---|---|
> | Baseline | - | 198.5 | 13.2 | 211.7 | 43.2 |
> | GUIDED with LLaMA-3.1-8B | 72.1 | 198.8 | 13.3 | 284.2 | 65.9 |
> | GUIDED with LLaMA-3.3-70B | 193.8 | 198.8 | 13.3 | 405.9 | 66.5 |
>
>
>
> **Table 4:** The Memory occupation(MB) of each module during inference in our GUIDED framework.
> | Method | LLM | Other |
> |---|---|---|
> | Baseline | - | 5277 |
> | GUIDED with LLaMA-3.1-8B | 6414 | 5329 |
> | GUIDED with LLaMA-3.3-70B | 44073 | 5329 |
>
>
>
>
>
>  **Failure cases.**
> We identify a key limitation when fine-grained attributes describe spatial relationships extending beyond the detected subject's bounding box. For example, for the query "suitcase on the conveyor belt", the conveyor belt (contextual attribute) may not be captured within the candidate box of any suitcase. Consequently, GUIDED may incorrectly associate the attribute with a suitcase spatially adjacent to the conveyor belt. This occurs because our method relies on the detector's candidate boxes, which may not encompass all required contextual elements. Due to the NIPS policy, we can not provide the visualization here, and we will include visualizations of such cases in the revised paper to clarify this failure mode.
>
> **Reproducibility concerns and the choice of LLMs.**
> We appreciate this important concern regarding reproducibility. To ensure transparency, we conducted experiments with open-source LLMs (`LLaMA-3.1-8B` and `LLaMA-3.3-70B`) as alternatives to `GPT-4o`. As quantified in Table 5, the mAP of GUIDED drops by merely 0.5\% with `LLaMA-3.1-8B` and improves by 0.1\% with `LLaMA-3.3-70B`. This demonstrates that GUIDED performance is not dependent on a specific proprietary model. We will provide more results with `LLaMA-3.1-8B` and `LLaMA-3.3-70B` forwith other OVD models and other datasets in our revised paper for reproducibility.
>
> **Table 5:** The comparison of mAP(%) evaluation results on the FG-OVD benchmark with different LLMs in subject identification.
> | **LLM** | **mAP** |
> |---|---|
> | GPT4-o | 66.4 |
> | LLaMA-3.1-8B | 65.9 |
> | LLaMA-3.3-70B | 66.5 |
>
>
>
>
> **Robustness of LLMs on subject identification.**
> To evaluate the robustness of LLMs on subject identification, we manually annotate the subjects from 300 fine-grained classes and assess the accuracy of subject identification with different LLMs. As summarized in Table~, failure cases are categorized into two types: (1) Hallucination toward in-context samples, where the LLM generates subjects irrelevant to the input text;  (2) Other errors, such as identifying an attribute instead of an object. Overall, all three LLMs achieve high correctness rates, demonstrating robust performance across diverse architectures.
> The results confirm the high robustness of this stage. Crucially, they show that while the smaller LLaMA-3.1-8B model is prone to hallucination errors that always propagate (8/8), this critical failure mode is completely eliminated by larger open-source models. Both LLaMA-3.3-70B and GPT-4o exhibit near-perfect performance, with their rare errors being minor and not always affecting the final detection. This demonstrates that our method's success is not tied to a specific proprietary model and is robust when using state-of-the-art open-source alternatives. We will add this analysis to our revised paper.
>
> **Table 6:** The number of failure cases in subject detection of 300 samples with different LLMs. The notation '$x/y$' denotes that there are $y$ failure cases, of which $x$ lead to detection errors.
>
> | LLM | Hallucination | Others | Total |
> |---|---|---|---|
> | GPT4-o | 0/0 | 1/2 | 1/2 |
> | LLaMA-3.1-8B | 8/8 | 1/1 | 9/9 |
> | LLaMA-3.3-70B | 0/0 | 1/1 | 1/1 |
>
>
> **Analysis of $m_{fine}$ and $m_{coarse}$.**
> $m_{fine}$ and $m_{coarse}$are used as temperatures to perform softmax with CLIP embeddings, and 50 is a common choice for the temperatures, which is extensively applied in previous work. We also provide the result with different $m_{\text{fine}}$ and $m_{\text{coarse}}$ in Table 7, performance remains robust across variations of these values.
>
>
> **Table 7:** Ablation study on $m_{fine}$ and $m_{coarse}$.
> | $m_{fine}$ | $m_{coarse}$ | MAP |
> |---|---|---|
> | 30 | 30 | 66.0 |
> | 50 | 50 | 66.4 |
> | 100 | 100 | 66.4 |

---

> > ### Comment · Reviewer_KZWx · 2025-08-05
> >
> > The author's response has addressed most of my concerns, but I still have a question regarding the ablation experiment for $\alpha$. From Table 1 in the rebuttal, the MAP metric is very sensitive to $\alpha$. It appears that using $s^{coarse}$ or $s^{fine}$ alone cannot produce high MAP, but setting $\alpha$ to $0.6$ reaches $66.4$ MAP (even higher than $17.1 + 28.0$). Is there an explanation for this? What is the MAP for other $\alpha$ values?

---

> > > ### Author Response · Authors · 2025-08-05
> > >
> > > We sincerely thank the reviewer for this insightful question.
> > > We first present the full ablation study on $\alpha$(also refer to the table 2 of our supplementary material). The results show the performance is robust within a reasonable range of $\alpha$.
> > > Specifically, when $\alpha = 0$, the scores rely exclusively on the attribute similarity scores from the refined CLIP, disregarding the crucial localization and coarse-grained object priors provided by the detector. When $\alpha=1$, the scores depend entirely on the initial confidence scores from the base detector. Since the base detector is modified to detect subjects by classifier embedding replacements, it lacks the fine-grained understanding to distinguish the attributes.
> > > When $\alpha$ is not set to 0 or 1, $ L_{\text{contrast}}$  is applied to the combined score. It trains the VLM's projection layer to adaptively scaling scores to balance contributions under different scale of $s^{\text{coarse}}$ and $s^{\text{fine}}$.  However, setting $\alpha = 0$ or $\alpha = 1$ eliminates the balance capability of $ L_{\text{contrast}}$ and leading to dramatic performance declines.
> > >
> > >
> > > **Table 1: The ablation of $\alpha$ on the FG-OVD dataset.**
> > >
> > > | $\alpha$ | Average |
> > > |:---:|:---:|
> > > | 1.0 | 28.0 |
> > > | 0.8 | 65.9 |
> > > | 0.6 | 66.4 |
> > > | 0.4 | 66.3 |
> > > | 0.2 | 65.3 |
> > > | 0.0 | 17.1 |
> > >
> > > We hope these detailed clarifications and additional results have resolved all ambiguities. We are grateful for the opportunity to improve our paper based on this valuable feedback.

---

> > > > ### Comment · Reviewer_KZWx · 2025-08-07
> > > >
> > > > All my concerns have been resolved. I will raise my score.

---

### Official Review · Reviewer_6ZvU · 2025-07-03

**Clarity:** 2
**Significance:** 3
**Originality:** 3
**Rating:** 5
**Confidence:** 3

**Summary:**

The paper proposes GUIDED, a method for fine-grained open-vocabulary detection (FG-OVD), a task aimed at detecting objects according to specific attribute-based text prompts. GUIDED decouples coarse-grained detection and fine-grained recognition and uses an LLM to identify the subject and attributes in the text prompt. Specifically, this framework is composed of "Subject Identification," where an LLM is used to decompose the text prompt into the subject and attributes, "Coarse-grained Object Detection," which combines a transformer-based object detector with attribute embedding fusion, and a "Fine-grained Attribute Discrimination Module" to identify region proposals from the detection stage that are aligned with the fine-grained attribute. GUIDED is evaluated on two FG-OVD datasets: FG-OVD and 3F-OVD. GUIDED achieves state of the art on these datasets. The authors report results applying GUIDED to the base models Grounding DINO, OWL-ViT, and LaMI-DETR, with the version based on LaMI-DETR performing the best overall.

**Questions:**

- What does "EDD" stand for in the tables? I assume it stands for GUIDED, but this needs to be clarified in the paper.
- Why is there a star next to HA-FGOVD in Table 1?
- What is the full inference time for a single example? This is important since the proposed framework includes an LLM, an extra (and potentially heavy) additional component compared to traditional detectors.

In light of the extensive experiments and good results, if the questions above can be clarified and integrated into the paper, then I would be willing to increase my score.

**Ethical Concerns:**

["NO or VERY MINOR ethics concerns only"]

**Final Justification:**

I have decided to raise my score since the authors have addressed my questions sufficiently. The authors have also mainly addressed the comments from the other reviewers. I hope the authors take the feedback from the discussion and improve the clarity and presentation of the final manuscript.

**Limitations:**

Yes

**Quality:**

2

**Strengths And Weaknesses:**

Strengths
- The proposed GUIDED method achieves state-of-the-art for fine-grained open-vocabulary detection
- The experiments are extensive, covering two different FG-OVD datasets and three different base models
- The authors present ablation studies testing the different components of the GUIDED framework

Weaknesses
- The key results are not presented in an easily accessible format. For example, the acronym "EDD" is not defined in the text even though it is included in many tables, and the text in Fig 1 (a) is difficult to read. The texts on the images in Fig 2 of the supplementary are also difficult to read.

---

> ### Author Rebuttal · Authors · 2025-07-30
>
> Thanks for the positive feedback on the extensive experiments and the superior performance of our method. We also appreciate the constructive comments to improve the paper. We would like to answer the proposed questions in the following:
>
>
> **The meaning of 'EDD'.**
> We apologize for the oversight. 'EDD' was a working name for our method from an earlier draft. We will correct this typo in the revised manuscript and perform a thorough check to ensure all terminology is consistent.
>
> **Texts in some figures are difficult to read.**
> We will resize and rearrange the text in these figures to improve clarity. Additionally, we will review all figures in the paper to ensure they are legible.
>
> **The meaning of the star next to HA-FGOVD.**
> As noted in the caption of Table 1 in our main paper, the star indicates that the results for HA-FGOVD are from our reproduction because the original paper did not provide these results. We will make this explanation more prominent in the caption to avoid any confusion.
>
> **Full inference time of a sample.**
> While GUIDED utilizes `GPT-4o` for its reasoning capabilities, measuring its API inference time directly is problematic as network latency becomes a confounding factor. To provide a meaningful assessment of the computational overhead introduced by our LLM component, we instead evaluated inference times using locally deployable LLMs: `LLaMA-3.1-8B` and `LLaMA-3.3-70B`. Crucially, these models achieve comparable mAP performance to `GPT-4o` on the FG-OVD task (as validated in Table 1), making them valid proxies for analyzing inference speed. We measured the inference time in Table 1 and will include these results in the revised paper.
>
> **Table 1:** The comparison of inference time(ms) between Baseline(LaMI-DETR) and GUIDED for detecting one class in an image. We also report the average mAP(%) of each method in FGOVD.
> | **Method** | LLM | Detector | CLIP | Overall | mAP |
> |---|---|---|---|---|---|
> | Baseline | - | 198.5 | 13.2 | 211.7 | 43.2 |
> | GUIDED with LLaMA-3.1-8B | 72.1 | 198.8 | 13.3 | 284.2 | 65.9 |
> | GUIDED with LLaMA-3.3-70B | 193.8 | 198.8 | 13.3 | 405.9 | 66.5 |

---

> > ### Comment · Reviewer_6ZvU · 2025-08-06
> > **Thank you for the clarifications**
> >
> > I would like to thank the authors for their response. I urge the authors to add these results and carefully check their final manuscript for typos and inconsistencies. That said, the authors have addressed my concerns, so I will raise my score accordingly.

---

### Note · Authors · 2025-08-16

We sincerely appreciate the reviewers' time and insightful feedback, particularly their positive recognition of our work's well motivations(KZWx,vfih), effective design(KZWx, vfih, HXfJ), strong performance (6ZvU, HXfJ) and extensive experiments(6ZvU, KZWx).
The valuable questions raised—particularly regarding computational overhead(6ZvU, KZWx, HXfJ), attribute fusion stability(vfih), reproducibility(KZWx), clarity(6ZvU, HXfJ), and OVD benchmark evaluations(vfih)—have prompted us to significantly strengthen the contributions and depth of this research through new experimental evidence.

To address vfih’s concerns, we clarify that the design choice of the AEF module and compare the AEF with simpler architectures. The results indicate the effectiveness of the AEF on attribute integrations. We also provide results on conventional OVD benchmarks (LVIS and MS-COCO) to show generalization and potential trade-offs of our proposed method.

For reproducibility and LLM robustness concerns(KZWx), we conduct experiments with different open-source LLMs. The improved mAP results and robust correctness rate demonstrate that open-source LLMs are competitive in subject identification, which confirms our approach's reproducibility.

Finally, regarding the computational overhead, we have provided the inference time analysis and memory occupation (addressing 6ZvU, KZWx and HXfJ) to illustrate the trade-offs between performance and inference efficiency.

All rebuttal insights, experiments, and clarifications will be incorporated into the revised manuscript. We believe these enhancements position our work to advance fine-grained open-vocabulary detection research and benefit the community.

---

### Decision · Program_Chairs · 2025-09-17

**Decision:**

Accept (poster)

**Comment:**

Overall, the paper introduces GUIDED, a well-motivated and systematic approach for fine-grained open-vocabulary object detection, addressing the key challenge of semantic entanglement between object categories and attributes. Reviewers highlighted the novelty of decomposing the task into subject identification, coarse-grained detection, and fine-grained attribute discrimination, along with the intuitive attribute embedding fusion module. Experiments are extensive, covering multiple FG-OVD benchmarks and base models, and the method achieves consistent state-of-the-art improvements. Some concerns were raised regarding accessibility of results, clarity in diagrams and variable definitions, computational overhead from LLM preprocessing, and reproducibility due to reliance on GPT-4o.

During the rebuttal, the authors provided clarifications, added missing results, and addressed technical concerns, resolving all reviewers’ questions. While minor limitations remain, such as additional LLM computation and broader benchmark evaluations, the core contributions, strong experimental validation, and novelty in addressing semantic entanglement convincingly support the quality and significance of the work. Based on the overall positive feedback and successful resolution of concerns, the final recommendation is to accept the paper.